# Universal Transformers

**Mostafa Dehghani**[*][†]
University of Amsterdam
dehghani@uva.nl

**Stephan Gouws**[*]
DeepMind
sgouws@google.com

**Oriol Vinyals**
DeepMind
vinyals@google.com

**Jakob Uszkoreit**
Google Brain
usz@google.com

**Łukasz Kaiser**
Google Brain
lukaszkaiser@google.com

## ABSTRACT

Recurrent neural networks (RNNs) sequentially process data by updating their state with each new data point, and have long been the de facto choice for sequence modeling tasks. However, their inherently sequential computation makes them slow to train. Feed-forward and convolutional architectures have recently been shown to achieve superior results on some sequence modeling tasks such as machine translation, with the added advantage that they concurrently process all inputs in the sequence, leading to easy parallelization and faster training times. Despite these successes, however, popular feed-forward sequence models like the Transformer fail to generalize in many simple tasks that recurrent models handle with ease, e.g. copying strings or even simple logical inference when the string or formula lengths exceed those observed at training time. We propose the Universal Transformer (UT), a parallel-in-time self-attentive recurrent sequence model which can be cast as a generalization of the Transformer model and which addresses these issues. UTs combine the parallelizability and global receptive field of feed-forward sequence models like the Transformer with the recurrent inductive bias of RNNs. We also add a dynamic per-position halting mechanism and find that it improves accuracy on several tasks. In contrast to the standard Transformer, under certain assumptions UTs can be shown to be Turing-complete. Our experiments show that UTs outperform standard Transformers on a wide range of algorithmic and language understanding tasks, including the challenging LAMBADA language modeling task where UTs achieve a new state of the art, and machine translation where UTs achieve a 0.9 BLEU improvement over Transformers on the WMT14 En-De dataset.

## 1 INTRODUCTION

Convolutional and fully-attentional feed-forward architectures like the Transformer have recently emerged as viable alternatives to recurrent neural networks (RNNs) for a range of sequence modeling tasks, notably machine translation (Gehring et al., 2017; Vaswani et al., 2017). These parallel-in-time architectures address a significant shortcoming of RNNs, namely their inherently sequential computation which prevents parallelization across elements of the input sequence, whilst still addressing the vanishing gradients problem as the sequence length gets longer (Hochreiter et al., 2003). The Transformer model in particular relies entirely on a self-attention mechanism (Parikh et al., 2016; Lin et al., 2017) to compute a series of context-informed vector-space representations of the symbols in its input and output, which are then used to predict distributions over subsequent symbols as the model predicts the output sequence symbol-by-symbol. Not only is this mechanism straightforward to parallelize, but as each symbol's representation is also directly informed by all other symbols' representations, this results in an effectively global receptive field across the whole sequence. This stands in contrast to e.g. convolutional architectures which typically only have a limited receptive field.

Notably, however, the Transformer with its fixed stack of distinct layers foregoes RNNs' inductive bias towards learning iterative or recursive transformations. Our experiments indicate that this inductive

---

[*] Equal contribution, alphabetically by last name.
[†] Work performed while at Google Brain.

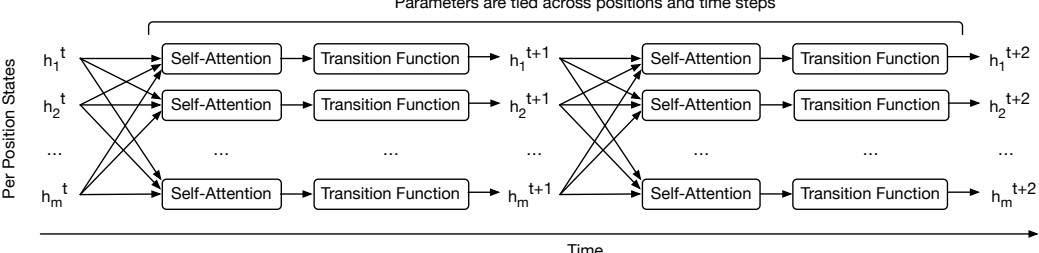

Figure 1: The Universal Transformer repeatedly refines a series of vector representations for each position of the sequence in parallel, by combining information from different positions using self-attention (see Eqn 2) and applying a recurrent transition function (see Eqn 4) across all time steps $1 \le t \le T$. We show this process over two recurrent time-steps. Arrows denote dependencies between operations. Initially, $h^0$ is initialized with the embedding for each symbol in the sequence. $h_i^t$ represents the representation for input symbol $1 \le i \le m$ at recurrent time-step $t$. With dynamic halting, $T$ is dynamically determined for each position (Section 2.2).

bias may be crucial for several algorithmic and language understanding tasks of varying complexity: in contrast to models such as the Neural Turing Machine (Graves et al., 2014), the Neural GPU (Kaiser & Sutskever, 2016) or Stack RNNs (Joulin & Mikolov, 2015), the Transformer does not generalize well to input lengths not encountered during training.

In this paper, we introduce the *Universal Transformer (UT)*, a parallel-in-time recurrent self-attentive sequence model which can be cast as a generalization of the Transformer model, yielding increased theoretical capabilities and improved results on a wide range of challenging sequence-to-sequence tasks. UTs combine the parallelizability and global receptive field of feed-forward sequence models like the Transformer with the recurrent inductive bias of RNNs, which seems to be better suited to a range of algorithmic and natural language understanding sequence-to-sequence problems. As the name implies, and in contrast to the standard Transformer, under certain assumptions UTs can be shown to be Turing-complete (or "computationally universal", as shown in Section 4).

In each recurrent step, the Universal Transformer iteratively refines its representations for all symbols in the sequence in parallel using a self-attention mechanism (Parikh et al., 2016; Lin et al., 2017), followed by a transformation (shared across all positions and time-steps) consisting of a depth-wise separable convolution (Chollet, 2016; Kaiser et al., 2017) or a position-wise fully-connected layer (see Fig 1). We also add a dynamic per-position halting mechanism (Graves, 2016), allowing the model to choose the required number of refinement steps *for each symbol* dynamically, and show for the first time that such a conditional computation mechanism can in fact improve accuracy on several smaller, structured algorithmic and linguistic inference tasks (although it marginally degraded results on MT).

Our strong experimental results show that UTs outperform Transformers and LSTMs across a wide range of tasks. The added recurrence yields improved results in machine translation where UTs outperform the standard Transformer. In experiments on several algorithmic tasks and the bAbI language understanding task, UTs also consistently and significantly improve over LSTMs and the standard Transformer. Furthermore, on the challenging LAMBADA text understanding data set UTs with dynamic halting achieve a new state of the art.

## 2 MODEL DESCRIPTION

### 2.1 THE UNIVERSAL TRANSFORMER

The Universal Transformer (UT; see Fig. 2) is based on the popular encoder-decoder architecture commonly used in most neural sequence-to-sequence models (Sutskever et al., 2014; Cho et al., 2014; Vaswani et al., 2017). Both the encoder and decoder of the UT operate by applying a recurrent neural network to the representations of each of the positions of the input and output sequence, respectively. However, in contrast to most applications of recurrent neural networks to sequential data, the UT does not recur over positions in the sequence, but over consecutive revisions of the vector representations of each position (i.e., over "depth"). In other words, the UT is not computationally bound by the number of symbols in the sequence, but only by the number of revisions made to each symbol's representation.

In each recurrent time-step, the representation of every position is concurrently (in parallel) revised in two sub-steps: first, using a self-attention mechanism to exchange information across all positions in the sequence, thereby generating a vector representation for each position that is informed by the representations of all other positions at the previous time-step. Then, by applying a transition function (shared across position and time) to the outputs of the self-attention mechanism, independently at each position. As the recurrent transition function can be applied any number of times, this implies that UTs can have variable depth (number of per-symbol processing steps). Crucially, this is in contrast to most popular neural sequence models, including the Transformer (Vaswani et al., 2017) or deep RNNs, which have constant depth as a result of applying a *fixed stack* of layers. We now describe the encoder and decoder in more detail.

**ENCODER:** Given an input sequence of length $m$, we start with a matrix whose rows are initialized as the $d$-dimensional embeddings of the symbols at each position of the sequence $H^0 \in \mathbb{R}^{m \times d}$. The UT then iteratively computes representations $H^t$ at step $t$ for all $m$ positions in parallel by applying the multi-headed dot-product self-attention mechanism from Vaswani et al. (2017), followed by a recurrent transition function. We also add residual connections around each of these function blocks and apply dropout and layer normalization (Srivastava et al., 2014; Ba et al., 2016) (see Fig. 2 for a simplified diagram, and Fig. 4 in the Appendix A for the complete model.).

More specifically, we use the scaled dot-product attention which combines queries $Q$, keys $K$ and values $V$ as follows

$$\text{ATTENTION}(Q, K, V) = \text{SOFTMAX}\left(\frac{QK^T}{\sqrt{d}}\right)V, \tag{1}$$

where $d$ is the number of columns of $Q$, $K$ and $V$. We use the multi-head version with $k$ heads, as introduced in (Vaswani et al., 2017),

$$\text{MULTIHEADSELFATTENTION}(H^t) = \text{CONCAT}(\text{head}_1, ..., \text{head}_k)W^O \tag{2}$$

$$\text{where head}_i = \text{ATTENTION}(H^t W_i^Q, H^t W_i^K, H^t W_i^V) \tag{3}$$

and we map the state $H^t$ to queries, keys and values with affine projections using learned parameter matrices $W^Q \in \mathbb{R}^{d \times d/k}$, $W^K \in \mathbb{R}^{d \times d/k}$, $W^V \in \mathbb{R}^{d \times d/k}$ and $W^O \in \mathbb{R}^{d \times d}$.

At step $t$, the UT then computes revised representations $H^t \in \mathbb{R}^{m \times d}$ for all $m$ input positions as follows

$$H^t = \text{LAYERNORM}(A^t + \text{TRANSITION}(A^t)) \tag{4}$$

$$\text{where } A^t = \text{LAYERNORM}((H^{t-1} + P^t) + \text{MULTIHEADSELFATTENTION}(H^{t-1} + P^t)), \tag{5}$$

where $\text{LAYERNORM}()$ is defined in Ba et al. (2016), and $\text{TRANSITION}()$ and $P^t$ are discussed below.

Depending on the task, we use one of two different transition functions: either a separable convolution (Chollet, 2016) or a fully-connected neural network that consists of a single rectified-linear activation function between two affine transformations, applied position-wise, i.e. individually to each row of $A^t$.

$P^t \in \mathbb{R}^{m \times d}$ above are fixed, constant, two-dimensional (position, time) *coordinate embeddings*, obtained by computing the sinusoidal position embedding vectors as defined in (Vaswani et al., 2017) for the positions $1 \leq i \leq m$ and the time-step $1 \leq t \leq T$ separately for each vector-dimension $1 \leq j \leq d$, and summing:

$$P^t_{i,2j} = \sin(i/10000^{2j/d}) + \sin(t/10000^{2j/d}) \tag{6}$$

$$P^t_{i,2j+1} = \cos(i/10000^{2j/d}) + \cos(t/10000^{2j/d}). \tag{7}$$

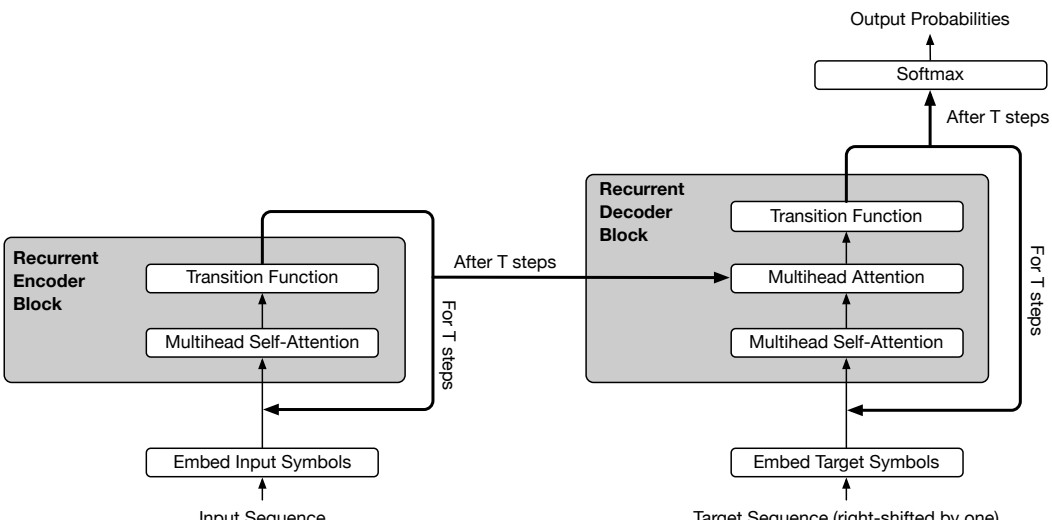

Figure 2: The recurrent blocks of the Universal Transformer encoder and decoder. This diagram omits position and time-step encodings as well as dropout, residual connections and layer normalization. A complete version can be found in Appendix A. The Universal Transformer with dynamic halting determines the number of steps $T$ for each position individually using ACT (Graves, 2016).

After $T$ steps (each updating all positions of the input sequence in parallel), the final output of the Universal Transformer encoder is a matrix of $d$-dimensional vector representations $H^T \in \mathbb{R}^{m \times d}$ for the $m$ symbols of the input sequence.

**DECODER:** The decoder shares the same basic recurrent structure of the encoder. However, after the self-attention function, the decoder additionally also attends to the final encoder representation $H^T$ of each position in the input sequence using the same multihead dot-product attention function from Equation 2, but with queries $Q$ obtained from projecting the decoder representations, and keys and values ($K$ and $V$) obtained from projecting the encoder representations (this process is akin to standard attention (Bahdanau et al., 2014)).

Like the Transformer model, the UT is autoregressive (Graves, 2013). Trained using teacher-forcing, at generation time it produces its output one symbol at a time, with the decoder consuming the previously produced output positions. During training, the decoder input is the target output, shifted to the right by one position. The decoder self-attention distributions are further masked so that the model can only attend to positions to the left of any predicted symbol. Finally, the per-symbol target distributions are obtained by applying an affine transformation $O \in \mathbb{R}^{d \times V}$ from the final decoder state to the output vocabulary size $V$, followed by a softmax which yields an $(m \times V)$-dimensional output matrix normalized over its rows:

$$p\big(y_{pos}|y_{[1:pos-1]},H^T\big) = \text{SOFTMAX}(OH^T)^1 \tag{8}$$

To generate from the model, the encoder is run once for the conditioning input sequence. Then the decoder is run repeatedly, consuming all already-generated symbols, while generating one additional distribution over the vocabulary for the symbol at the next output position per iteration. We then typically sample or select the highest probability symbol as the next symbol.

## 2.2 DYNAMIC HALTING

In sequence processing systems, certain symbols (e.g. some words or phonemes) are usually more ambiguous than others. It is therefore reasonable to allocate more processing resources to these more ambiguous symbols. Adaptive Computation Time (ACT) (Graves, 2016) is a mechanism for dynamically modulating the number of computational steps needed to process each input symbol

---

[1]Note that $T$ here denotes time-step $T$ and not the transpose operation.

| Model | 10K examples | | 1K examples | |
|---|---|---|---|---|
| | train single | train joint | train single | train joint |
| **Previous best results:** | | | | |
| QRNet (Seo et al., 2016) | 0.3 (0/20) | - | - | - |
| Sparse DNC (Rae et al., 2016) | - | 2.9 (1/20) | - | - |
| GA+MAGE Dhingra et al. (2017) | - | - | 8.7 (5/20) | - |
| MemN2N Sukhbaatar et al. (2015) | - | - | - | 12.4 (11/20) |
| **Our Results:** | | | | |
| Transformer (Vaswani et al., 2017) | 15.2 (10/20) | 22.1 (12/20) | 21.8 (5/20) | 26.8 (14/20) |
| Universal Transformer (this work) | 0.23 (0/20) | 0.47 (0/20) | 5.31 (5/20) | 8.50 (8/20) |
| UT w/ dynamic halting (this work) | **0.21 (0/20)** | **0.29 (0/20)** | **4.55 (3/20)** | **7.78 (5/20)** |

Table 1: Average error and number of failed tasks ($> 5\%$ error) out of 20 (in parentheses; lower is better in both cases) on the bAbI dataset under the different training/evaluation setups. We indicate state-of-the-art where available for each, or '-' otherwise.

(called the "ponder time") in standard recurrent neural networks based on a scalar *halting probability* predicted by the model at each step.

Inspired by the interpretation of Universal Transformers as applying self-attentive RNNs in parallel to all positions in the sequence, we also add a dynamic ACT halting mechanism to each position (i.e. to each per-symbol self-attentive RNN; see Appendix C for more details). Once the per-symbol recurrent block halts, its state is simply copied to the next step until all blocks halt, or we reach a maximum number of steps. The final output of the encoder is then the final layer of representations produced in this way.

## 3 EXPERIMENTS AND ANALYSIS

We evaluated the Universal Transformer on a range of algorithmic and language understanding tasks, as well as on machine translation. We describe these tasks and datasets in more detail in Appendix D.

### 3.1 BABI QUESTION-ANSWERING

The bAbi question answering dataset (Weston et al., 2015) consists of 20 different tasks, where the goal is to answer a question given a number of English sentences that encode potentially multiple supporting facts. The goal is to measure various forms of language understanding by requiring a certain type of reasoning over the linguistic facts presented in each story. A standard Transformer does not achieve good results on this task[2]. However, we have designed a model based on the Universal Transformer which achieves state-of-the-art results on this task.

To encode the input, similar to Henaff et al. (2016), we first encode each fact in the story by applying a learned multiplicative positional mask to each word's embedding, and summing up all embeddings. We embed the question in the same way, and then feed the (Universal) Transformer with these embeddings of the facts and questions.

As originally proposed, models can either be trained on each task separately ("train single") or jointly on all tasks ("train joint"). Table 1 summarizes our results. We conducted 10 runs with different initializations and picked the best model based on performance on the validation set, similar to previous work. Both the UT and UT with dynamic halting achieve state-of-the-art results on all tasks in terms of average error and number of failed tasks[3], in both the 10K and 1K training regime (see Appendix E for breakdown by task).

To understand the working of the model better, we analyzed both the attention distributions and the average ACT ponder times for this task (see Appendix F for details). First, we observe that the attention distributions start out very uniform, but get progressively sharper in later steps around the correct supporting facts that are required to answer each question, which is indeed very similar to how humans would solve the task. Second, with dynamic halting we observe that the average ponder time (i.e. depth

---

[2]We experimented with different hyper-parameters and different network sizes, but it always overfits.

[3]Defined as $> 5\%$ error.

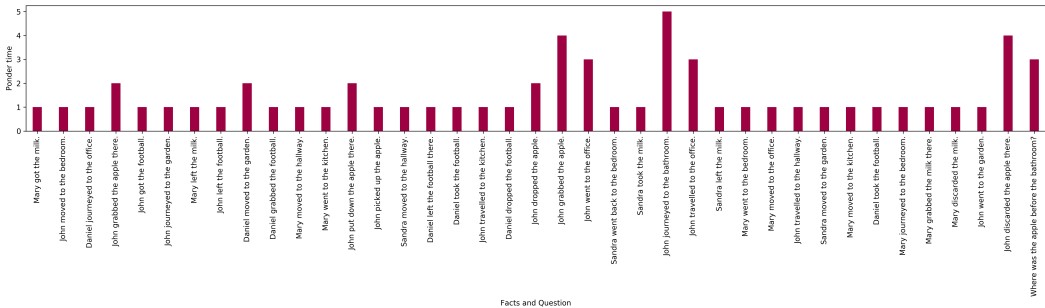

Figure 3: Ponder time of UT with dynamic halting for encoding facts in a story and question in a bAbI task requiring three supporting facts.

of the per-symbol recurrent processing chain) over all positions in all samples in the test data for tasks requiring three supporting facts is higher (3.8±2.2) than for tasks requiring only two (3.1±1.1), which is in turn higher than for tasks requiring only one supporting fact (2.3±0.8). This indicates that the model adjusts the number of processing steps with the number of supporting facts required to answer the questions. Finally, we observe that the histogram of ponder times at different positions is more uniform in tasks requiring only one supporting fact compared to two and three, and likewise for tasks requiring two compared to three. Especially for tasks requiring three supporting facts, many positions halt at step 1 or 2 already and only a few get transformed for more steps (see for example Fig 3). This is particularly interesting as the length of stories is indeed much higher in this setting, with more irrelevant facts which the model seems to successfully learn to ignore in this way.

Similar to dynamic memory networks (Kumar et al., 2016), there is an iterative attention process in UTs that allows the model to condition its attention over memory on the result of previous iterations. Appendix F presents some examples illustrating that there is a notion of temporal states in UT, where the model updates its states (memory) in each step based on the output of previous steps, and this chain of updates can also be viewed as steps in a multi-hop reasoning process.

## 3.2    SUBJECT-VERB AGREEMENT

Next, we consider the task of predicting number-agreement between subjects and verbs in English sentences (Linzen et al., 2016). This task acts as a proxy for measuring the ability of a model to capture hierarchical (dependency) structure in natural language sentences. We use the dataset provided by (Linzen et al., 2016) and follow their experimental protocol of solving the task using a language modeling training setup, i.e. a next word prediction objective, followed by calculating the ranking accuracy of the target verb at test time. We evaluated our model on subsets of the test data with different task difficulty, measured in terms of *agreement attractors* – the number of intervening nouns with the opposite number from the subject (meant to confuse the model). For example, given the sentence *The keys to the cabinet*[4], the objective during training is to predict the verb *are* (plural). At test time, we then evaluate the ranking accuracy of the agreement attractors: i.e. the goal is to rank *are* higher than *is* in this case.

Our results are summarized in Table 2. The best LSTM with attention from the literature achieves 99.18% on this task (Yogatama et al., 2018), outperforming a vanilla Transformer (Tran et al., 2018). UTs significantly outperform standard Transformers, and achieve an *average* result comparable to the current state of the art (99.2%). However, we see that UTs (and particularly with dynamic halting) perform progressively better than all other models as the number of attractors increases (see the last row, $\Delta$).

## 3.3    LAMBADA LANGUAGE MODELING

The LAMBADA task (Paperno et al., 2016) is a language modeling task consisting of predicting a missing target word given a broader context of 4-5 preceding sentences. The dataset was specifically designed so that humans are able to accurately predict the target word when shown the full context, but not when only shown the target sentence in which it appears. It therefore goes beyond language

---

[4]*Cabinet* (singular) is an agreement attractor in this case.

| Model | Number of attractors | | | | | | |
|---|---|---|---|---|---|---|---|
| | 0 | 1 | 2 | 3 | 4 | 5 | Total |
| **Previous best results (Yogatama et al., 2018):** | | | | | | | |
| Best Stack-RNN | *0.994* | 0.979 | 0.965 | 0.935 | 0.916 | 0.880 | 0.992 |
| Best LSTM | 0.993 | 0.972 | 0.950 | 0.922 | 0.900 | 0.842 | 0.991 |
| Best Attention | **0.994** | **0.977** | 0.959 | 0.929 | 0.907 | 0.842 | **0.992** |
| **Our results:** | | | | | | | |
| Transformer | 0.973 | 0.941 | 0.932 | 0.917 | 0.901 | 0.883 | 0.962 |
| Universal Transformer | 0.993 | 0.971 | **0.969** | 0.940 | 0.921 | 0.892 | **0.992** |
| UT w/ ACT | **0.994** | 0.969 | 0.967 | **0.944** | **0.932** | **0.907** | **0.992** |
| Δ (UT w/ ACT - Best) | 0 | -0.008 | 0.002 | 0.009 | 0.016 | 0.027 | - |

Table 2: Accuracy on the subject-verb agreement number prediction task (higher is better).

| Model | LM Perplexity & (Accuracy) | | | RC Accuracy | | |
|---|---|---|---|---|---|---|
| | control | dev | test | control | dev | test |
| Neural Cache (Grave et al., 2016) | **129** | 139 | - | - | - | - |
| Dhingra et al. Dhingra et al. (2018) | - | - | - | - | - | 0.5569 |
| Transformer | 142 (0.19) | 5122 (0.0) | 7321 (0.0) | 0.4102 | 0.4401 | 0.3988 |
| LSTM | 138 (0.23) | 4966 (0.0) | 5174 (0.0) | 0.1103 | 0.2316 | 0.2007 |
| UT *base*, 6 steps (fixed) | 131 (0.32) | 279 (0.18) | 319 (0.17) | **0.4801** | 0.5422 | 0.5216 |
| UT w/ dynamic halting | 130 (0.32) | **134** (0.22) | **142** (0.19) | 0.4603 | **0.5831** | **0.5625** |
| UT *base*, 8 steps (fixed) | 129(0.32) | 192 (0.21) | 202 (0.18) | - | - | - |
| UT *base*, 9 steps (fixed) | **129(0.33)** | 214 (0.21) | 239 (0.17) | - | - | - |

Table 3: LAMBADA language modeling (LM) perplexity (lower better) with accuracy in parentheses (higher better), and Reading Comprehension (RC) accuracy results (higher better). '-' indicates no reported results in that setting.

modeling, and tests the ability of a model to incorporate broader discourse and longer term context when predicting the target word.

The task is evaluated in two settings: as *language modeling* (the standard setup) and as *reading comprehension*. In the former (more challenging) case, a model is simply trained for next-word prediction on the training data, and evaluated on the target words at test time (i.e. the model is trained to predict all words, not specifically challenging target words). In the latter setting, introduced by Chu et al. Chu et al. (2017), the target sentence (minus the last word) is used as query for selecting the target word from the context sentences. Note that the target word appears in the context 81% of the time, making this setup much simpler. However the task is impossible in the remaining 19% of the cases.

The results are shown in Table 3. Universal Transformer achieves state-of-the-art results in both the language modeling and reading comprehension setup, outperforming both LSTMs and vanilla Transformers. Note that the control set was constructed similar to the LAMBADA development and test sets, but without filtering them in any way, so achieving good results on this set shows a model's strength in standard language modeling.

Our best fixed UT results used 6 steps. However, the average number of steps that the best UT with dynamic halting took on the test data over all positions and examples was $8.2\pm2.1$. In order to see if the dynamic model did better simply because it took more steps, we trained two fixed UT models with 8 and 9 steps respectively (see last two rows). Interestingly, these two models achieve better results compared to the model with 6 steps, but *do not outperform the UT with dynamic halting*. This leads us to believe that dynamic halting may act as a useful regularizer for the model via incentivizing a smaller numbers of steps for some of the input symbols, while allowing more computation for others.

## 3.4 ALGORITHMIC TASKS

We trained UTs on three algorithmic tasks, namely Copy, Reverse, and (integer) Addition, all on strings composed of decimal symbols ('0'-'9'). In all the experiments, we train the models on sequences of length 40 and evaluated on sequences of length 400 (Kaiser & Sutskever, 2016). We

| Model | Copy | | Reverse | | Addition | |
|---|---|---|---|---|---|---|
| | char-acc | seq-acc | char-acc | seq-acc | char-acc | seq-acc |
| LSTM | 0.45 | 0.09 | 0.66 | 0.11 | 0.08 | 0.0 |
| Transformer | 0.53 | 0.03 | 0.13 | 0.06 | 0.07 | 0.0 |
| Universal Transformer | 0.91 | 0.35 | 0.96 | 0.46 | 0.34 | 0.02 |
| Neural GPU* | **1.0** | **1.0** | **1.0** | **1.0** | **1.0** | **1.0** |

Table 4: Accuracy (higher better) on the algorithmic tasks. *Note that the Neural GPU was trained with a special curriculum to obtain the perfect result, while other models are trained without any curriculum.

| | Copy | | Double | | Reverse | |
|---|---|---|---|---|---|---|
| Model | char-acc | seq-acc | char-acc | seq-acc | char-acc | seq-acc |
| LSTM | 0.78 | 0.11 | 0.51 | 0.047 | 0.91 | 0.32 |
| Transformer | 0.98 | 0.63 | 0.94 | 0.55 | 0.81 | 0.26 |
| Universal Transformer | **1.0** | **1.0** | **1.0** | **1.0** | **1.0** | **1.0** |

Table 5: Character-level (*char-acc*) and sequence-level accuracy (*seq-acc*) results on the Memorization LTE tasks, with maximum length of 55.

| | Program | | Control | | Addition | |
|---|---|---|---|---|---|---|
| Model | char-acc | seq-acc | char-acc | seq-acc | char-acc | seq-acc |
| LSTM | 0.53 | 0.12 | 0.68 | 0.21 | 0.83 | 0.11 |
| Transformer | 0.71 | 0.29 | 0.93 | 0.66 | **1.0** | **1.0** |
| Universal Transformer | **0.89** | **0.63** | **1.0** | **1.0** | **1.0** | **1.0** |

Table 6: Character-level (*char-acc*) and sequence-level accuracy (*seq-acc*) results on the Program Evaluation LTE tasks with maximum nesting of 2 and length of 5.

train UTs using positions starting with randomized offsets to further encourage the model to learn position-relative transformations. Results are shown in Table 4. The UT outperforms both LSTM and vanilla Transformer by a wide margin on all three tasks. The Neural GPU reports perfect results on this task (Kaiser & Sutskever, 2016), however we note that this result required a special curriculum-based training protocol which was not used for other models.

## 3.5 LEARNING TO EXECUTE (LTE)

As another class of sequence-to-sequence learning problems, we also evaluate UTs on tasks indicating the ability of a model to learn to execute computer programs, as proposed in (Zaremba & Sutskever, 2015). These tasks include program evaluation tasks (program, control, and addition), and memorization tasks (copy, double, and reverse).

We use the mix-strategy discussed in (Zaremba & Sutskever, 2015) to generate the datasets. Unlike (Zaremba & Sutskever, 2015), we do not use any curriculum learning strategy during training and we make no use of target sequences at test time. Tables 5 and 6 present the performance of an LSTM model, Transformer, and Universal Transformer on the program evaluation and memorization tasks, respectively. UT achieves perfect scores in all the memorization tasks and also outperforms both LSTMs and Transformers in all program evaluation tasks by a wide margin.

## 3.6 MACHINE TRANSLATION

We trained a UT on the WMT 2014 English-German translation task using the same setup as reported in (Vaswani et al., 2017) in order to evaluate its performance on a large-scale sequence-to-sequence task. Results are summarized in Table 7. The UT with a fully-connected recurrent transition function (instead of separable convolution) and without ACT improves by 0.9 BLEU over a Transformer and 0.5 BLEU over a Weighted Transformer with approximately the same number of parameters (Ahmed et al., 2017).

| Model | BLEU |
|---|---|
| Universal Transformer *small* | 26.8 |
| Transformer *base* (Vaswani et al., 2017) | 28.0 |
| Weighted Transformer *base* (Ahmed et al., 2017) | 28.4 |
| Universal Transformer *base* | **28.9** |

Table 7: Machine translation results on the WMT14 En-De translation task trained on 8xP100 GPUs in comparable training setups. All *base* results have the same number of parameters.

## 4 DISCUSSION

When running for a fixed number of steps, the Universal Transformer is equivalent to a multi-layer Transformer with tied parameters across all its layers. This is partly similar to the Recursive Transformer, which ties the weights of its self-attention layers across depth (Gulcehre et al., 2018)[5]. However, as the per-symbol recurrent transition functions can be applied any number of times, another and possibly more informative way of characterizing the UT is as a block of parallel RNNs (one for each symbol, with shared parameters) evolving per-symbol hidden states concurrently, generated at each step by attending to the sequence of hidden states at the previous step. In this way, it is related to architectures such as the Neural GPU (Kaiser & Sutskever, 2016) and the Neural Turing Machine (Graves et al., 2014). UTs thereby retain the attractive computational efficiency of the original feed-forward Transformer model, but with the added recurrent inductive bias of RNNs. Furthermore, using a dynamic halting mechanism, UTs can choose the number of processing steps based on the input data.

The connection between the Universal Transformer and other sequence models is apparent from the architecture: if we limited the recurrent steps to one, it would be a Transformer. But it is more interesting to consider the relationship between the Universal Transformer and RNNs and other networks where recurrence happens over the time dimension. Superficially these models may seem closely related since they are recurrent as well. But there is a crucial difference: time-recurrent models like RNNs cannot access memory in the recurrent steps. This makes them computationally more similar to automata, since the only memory available in the recurrent part is a fixed-size state vector. UTs on the other hand can attend to the whole previous layer, allowing it to access memory in the recurrent step.

Given sufficient memory the Universal Transformer is computationally universal – i.e. it belongs to the class of models that can be used to simulate any Turing machine, thereby addressing a shortcoming of the standard Transformer model [6]. In addition to being theoretically appealing, our results show that this added expressivity also leads to improved accuracy on several challenging sequence modeling tasks. This closes the gap between practical sequence models competitive on large-scale tasks such as machine translation, and computationally universal models such as the Neural Turing Machine or the Neural GPU (Graves et al., 2014; Kaiser & Sutskever, 2016), which can be trained using gradient descent to perform algorithmic tasks.

To show this, we can reduce a Neural GPU to a Universal Transformer. Ignoring the decoder and parameterizing the self-attention module, i.e. self-attention with the residual connection, to be the identity function, we assume the transition function to be a convolution. If we now set the total number of recurrent steps $T$ to be equal to the input length, we obtain exactly a Neural GPU. Note that the last step is where the Universal Transformer crucially differs from the vanilla Transformer whose depth cannot scale dynamically with the size of the input. A similar relationship exists between the Universal Transformer and the Neural Turing Machine, whose single read/write operations per step can be expressed by the global, parallel representation revisions of the Universal Transformer. In contrast to these models, however, which only perform well on algorithmic tasks, the Universal Transformer also achieves competitive results on realistic natural language tasks such as LAMBADA and machine translation.

Another related model architecture is that of end-to-end Memory Networks (Sukhbaatar et al., 2015). In contrast to end-to-end memory networks, however, the Universal Transformer uses memory corresponding to states aligned to individual positions of its inputs or outputs. Furthermore, the Universal Transformer follows the encoder-decoder configuration and achieves competitive performance in large-scale sequence-to-sequence tasks.

---

[5]Note that in UT both the self-attention and transition weights are tied across layers.

[6]Appendix B illustrates how UT is computationally more powerful than the standard Transformer.

## 5 CONCLUSION

This paper introduces the Universal Transformer, a generalization of the Transformer model that extends its theoretical capabilities and produces state-of-the-art results on a wide range of challenging sequence modeling tasks, such as language understanding but also a variety of algorithmic tasks, thereby addressing a key shortcoming of the standard Transformer. The Universal Transformer combines the following key properties into one model:

**Weight sharing**: Following intuitions behind weight sharing found in CNNs and RNNs, we extend the Transformer with a simple form of weight sharing that strikes an effective balance between inductive bias and model expressivity, which we show extensively on both small and large-scale experiments.

**Conditional computation**: In our goal to build a computationally universal machine, we equipped the Universal Transformer with the ability to halt or continue computation through a recently introduced mechanism, which shows stronger results compared to the fixed-depth Universal Transformer.

We are enthusiastic about the recent developments on parallel-in-time sequence models. By adding computational capacity and recurrence in processing depth, we hope that further improvements beyond the basic Universal Transformer presented here will help us build learning algorithms that are both more powerful, data efficient, and generalize beyond the current state-of-the-art.

The code used to train and evaluate Universal Transformers is available at `https://github.com/tensorflow/tensor2tensor` (Vaswani et al., 2018).

**Acknowledgements**   We are grateful to Ashish Vaswani, Douglas Eck, and David Dohan for their fruitful comments and inspiration.

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

## APPENDIX A   DETAILED SCHEMA OF THE UNIVERSAL TRANSFORMER

Figure 4: The Universal Transformer with position and step embeddings as well as dropout and layer normalization.

## APPENDIX B   ON THE COMPUTATIONAL POWER OF UT VS TRANSFORMER

With respect to their computational power, the key difference between the Transformer and the Universal Transformer lies in the number of sequential steps of computation (i.e. in depth). While a standard Transformer executes a total number of operations that scales with the input size, the number of sequential operations is constant, independent of the input size and determined solely by the number of layers. Assuming finite precision, this property implies that the standard Transformer cannot be computationally universal. When choosing a number of steps as a function of the input length, however, the Universal Transformer does not suffer from this limitation. Note that this holds independently of whether or not adaptive computation time is employed but does assume a non-constant, even if possibly deterministic, number of steps. Varying the number of steps dynamically after training is enabled by sharing weights across sequential computation steps in the Universal Transformer.

An intuitive example are functions whose execution requires the sequential processing of each input element. In this case, for any given choice of depth $T$, one can construct an input sequence of length $N > T$ that cannot be processed correctly by a standard Transformer. With an appropriate, input-length dependent choice of sequential steps, however, a Universal Transformer, RNNs or Neural GPUs can execute such a function.

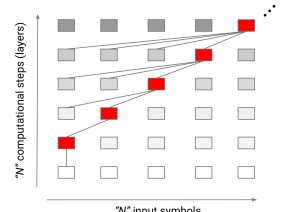

## APPENDIX C   UT WITH DYNAMIC HALTING

We implement the dynamic halting based on ACT (Graves, 2016) as follows in TensorFlow. In each step of the UT with dynamic halting, we are given the halting probabilities, remainders, number of updates up to that point, and the previous state (all initialized as zeros), as well as a scalar threshold between 0 and 1 (a hyper-parameter). We then compute the new state for each position and calculate the new per-position halting probabilities based on the state for each position. The UT then decides to halt for some positions that crossed the threshold, and updates the state of other positions until the model halts for all positions or reaches a predefined maximum number of steps:

```python
# While-loop stops when this predicate is FALSE
# i.e. all (( probability < threshold) & (counter < max_steps)) are false
def should_continue(u0, u1, halting_probability, u2, n_updates, u3):
    return tf.reduce_any(
            tf.logical_and(
                tf.less(halting_probability, threshold),
                tf.less(n_updates, max_steps)))
# Do while loop iterations until predicate above is false
(_, _, _, remainder, n_updates, new_state) = tf.while_loop(
    should_continue, ut_with_dynamic_halting, (state,
    step, halting_probability, remainders, n_updates, previous_state))
```

Listing 1: UT with dynamic halting.

The following shows the computations in each step:

```python
def ut_with_dynamic_halting(state, step, halting_probability,
                            remainders, n_updates, previous_state):
    # Calculate the probabilities based on the state
    p = common_layers.dense(state, 1, activation=tf.nn.sigmoid,
            use_bias=True)
    # Mask for inputs which have not halted yet
    still_running = tf.cast(
        tf.less(halting_probability ,1.0), tf.float32)
    # Mask of inputs which halted at this step
    new_halted = tf.cast(
        tf.greater(halting_probability + p * still_running, threshold),
            tf.float32) * still_running
    # Mask of inputs which haven't halted, and didn't halt this step
    still_running = tf.cast(
        tf.less_equal(halting_probability + p * still_running,
            threshold), tf.float32) * still_running
    # Add the halting probability for this step to the halting
    # probabilities for those inputs which haven't halted yet
    halting_probability += p * still_running
    # Compute remainders for the inputs which halted at this step
    remainders += new_halted * (1 - halting_probability)
    # Add the remainders to those inputs which halted at this step
    halting_probability += new_halted * remainders
    # Increment n_updates for all inputs which are still running
    n_updates += still_running + new_halted
    # Compute the weight to be applied to the new state and output:
    #   0 when the input has already halted,
    #   p when the input hasn't halted yet,
    #   the remainders when it halted this step.
    update_weights = tf.expand_dims(p * still_running +
                                    new_halted * remainders, -1)
    # Apply transformation to the state
    transformed_state = transition_function(self_attention(state))
    # Interpolate transformed and previous states for non-halted inputs
    new_state = ((transformed_state * update_weights) +
                (previous_state * (1 - update_weights)))
    step += 1
    return (transformed_state, step, halting_probability,
            remainders, n_updates, new_state)
```

Listing 2: Computations in each step of the UT with dynamic halting.

## APPENDIX D  DESCRIPTION OF SOME OF THE TASKS/DATASETS

Here, we provide some additional details on the bAbI, subject-verb agreement, LAMBADA language modeling, and learning to execute (LTE) tasks.

### D.1  BABI QUESTION-ANSWERING

The bAbi question answering dataset (Weston et al., 2015) consists of 20 different synthetic tasks[7]. The aim is that each task tests a unique aspect of language understanding and reasoning, including the ability of: reasoning from supporting facts in a story, answering true/false type questions, counting, understanding negation and indefinite knowledge, understanding coreferences, time reasoning, positional and size reasoning, path-finding, and understanding motivations (to see examples for each of these tasks, please refer to Table 1 in (Weston et al., 2015)).

There are two versions of the dataset, one with 1k training examples and the other with 10k examples. It is important for a model to be data-efficient to achieve good results using only the 1k training examples. Moreover, the original idea is that a single model should be evaluated across all the tasks (not tuning per task), which is the *train joint* setup in Table 1, and the tables presented in Appendix E.

### D.2  SUBJECT-VERB AGREEMENT

Subject-verb agreement is the task of predicting number agreement between subject and verb in English sentences. Succeeding in this task is a strong indicator that a model can learn to approximate syntactic structure and therefore it was proposed by Linzen et al. (2016) as proxy for assessing the ability of different models to capture hierarchical structure in natural language.

Two experimental setups were proposed by Linzen et al. (2016) for training a model on this task: 1) training with a language modeling objective, i.e., next word prediction, and 2) as binary classification, i.e. predicting the number of the verb given the sentence. In this paper, we use the language modeling objective, meaning that we provide the model with an implicit supervision and evaluate based on the ranking accuracy of the correct form of the verb compared to the incorrect form of the verb.

In this task, in order to have different levels of difficulty, "agreement attractors" are used, i.e. one or more intervening nouns with the opposite number from the subject with the goal of confusing the model. In this case, the model needs to correctly identify the head of the syntactic subject that corresponds to a given verb and ignore the intervening attractors in order to predict the correct form of that verb. Here are some examples for this task in which subjects and the corresponding verbs are in boldface and agreement attractors are underlined:

```
No attractor:       The boy smiles.
One attractor:      The number of men is not clear.
Two attractors:     The ratio of men to women is not clear.
Three attractors:   The ratio of men to women and children is not clear.
```

### D.3  LAMBADA LANGUAGE MODELING

The LAMBADA task (Paperno et al., 2016) is a broad context language modeling task. In this task, given a narrative passage, the goal is to predict the last word (target word) of the last sentence (target sentence) in the passage. These passages are specifically selected in a way that human subjects are easily able to guess their last word if they are exposed to a long passage, but not if they only see the target sentence preceding the target word[8]. Here is a sample from the dataset:

```
Context:
                 "Yes, I thought I was going to lose the baby."
                 "I was scared too," he stated, sincerity flooding his eyes.
                 "You were?"  "Yes, of course.  Why do you even ask?"
                 "This baby wasn't exactly planned for."
Target sentence:
                 "Do you honestly think that I would want you to have a ________?"
Target word:
                 miscarriage
```

The LAMBADA task consists in predicting the target word given the whole passage (i.e., the context plus the target sentence). A "control set" is also provided which was constructed by randomly sampling passages of the same shape and size as the ones used to build LAMBADA, but without filtering them in any way. The control

---

[7]https://research.fb.com/downloads/babi
[8]http://clic.cimec.unitn.it/lambada/appendix_onefile.pdf

set is used to evaluate the models at standard language modeling before testing on the LAMBADA task, and therefore to ensure that low performance on the latter cannot be attributed simply to poor language modeling.

The task is evaluated in two settings: as *language modeling* (the standard setup) and as *reading comprehension*. In the former (more challenging) case, a model is simply trained for the next word prediction on the training data, and evaluated on the target words at test time (i.e. the model is trained to predict all words, not specifically challenging target words). In this paper, we report the results of the Universal Transformer in both setups.

## D.4  LEARNING TO EXECUTE (LTE)

LTE is a set of tasks indicating the ability of a model to learn to execute computer programs and was proposed by Zaremba & Sutskever (2015). These tasks include two subsets: 1) program evaluation tasks (program, control, and addition) that are designed to assess the ability of models for understanding numerical operations, if-statements, variable assignments, the compositionality of operations, and more, as well as 2) memorization tasks (copy, double, and reverse).

The difficulty of the program evaluation tasks is parameterized by their *length* and *nesting*. The length parameter is the number of digits in the integers that appear in the programs (so the integers are chosen uniformly from [1, *length*]), and the nesting parameter is the number of times we are allowed to combine the operations with each other. Higher values of nesting yield programs with deeper parse trees. For instance, here is a program that is generated with length = 4 and nesting = 3.

```
Input:
        j=8584
        for x in range(8):
          j+=920
        b=(1500+j)
        print((b+7567))
Target:
        25011
```

# APPENDIX E   BABI DETAILED RESULTS

| Task id | Best seed run for each task (out of 10 runs) | | | |
|---|---|---|---|---|
| | 10K | | 1K | |
| | train single | train joint | train single | train joint |
| 1 | 0.0 | 0.0 | 0.0 | 0.0 |
| 2 | 0.0 | 0.0 | 0.0 | 0.5 |
| 3 | 0.4 | 1.2 | 3.7 | 5.4 |
| 4 | 0.0 | 0.0 | 0.0 | 0.0 |
| 5 | 0.0 | 0.0 | 0.0 | 0.5 |
| 6 | 0.0 | 0.0 | 0.0 | 0.5 |
| 7 | 0.0 | 0.0 | 0.0 | 3.2 |
| 8 | 0.0 | 0.0 | 0.0 | 1.6 |
| 9 | 0.0 | 0.0 | 0.0 | 0.2 |
| 10 | 0.0 | 0.0 | 0.0 | 0.4 |
| 11 | 0.0 | 0.0 | 0.0 | 0.1 |
| 12 | 0.0 | 0.0 | 0.0 | 0.0 |
| 13 | 0.0 | 0.0 | 0.0 | 0.6 |
| 14 | 0.0 | 0.0 | 0.0 | 3.8 |
| 15 | 0.0 | 0.0 | 0.0 | 5.9 |
| 16 | 0.4 | 1.2 | 5.8 | 15.4 |
| 17 | 0.6 | 0.2 | 32.0 | 42.9 |
| 18 | 0.0 | 0.0 | 0.0 | 4.1 |
| 19 | 2.8 | 3.1 | 47.1 | 68.2 |
| 20 | 0.0 | 0.0 | 2.4 | 2.4 |
| avg err | 0.21 | 0.29 | 4.55 | 7.78 |
| failed | 0 | 0 | 3 | 5 |

| Task id | Average (±var) over all seeds (for 10 runs) | | | |
|---|---|---|---|---|
| | 10K | | 1K | |
| | train single | train joint | train single | train joint |
| 1 | 0.0 ±0.0 | 0.0 ±0.0 | 0.2 ±0.3 | 0.1 ±0.2 |
| 2 | 0.2 ±0.4 | 1.7 ±2.6 | 3.2 ±4.1 | 4.3 ±11.6 |
| 3 | 1.8 ±1.8 | 4.6 ±7.3 | 9.1 ±12.7 | 14.3 ±18.1 |
| 4 | 0.1 ±0.1 | 0.2 ±0.1 | 0.3 ±0.3 | 0.4 ±0.6 |
| 5 | 0.2 ±0.3 | 0.8 ±0.5 | 1.1 ±1.3 | 4.3 ±5.6 |
| 6 | 0.1 ±0.2 | 0.1 ±0.2 | 1.2 ±2.1 | 0.8 ±0.4 |
| 7 | 0.3 ±0.5 | 1.1 ±1.5 | 0.0 ±0.0 | 4.1 ±2.9 |
| 8 | 0.3 ±0.2 | 0.5 ±1.1 | 0.1 ±0.2 | 3.9 ±4.2 |
| 9 | 0.0 ±0.0 | 0.0 ±0.0 | 0.1 ±0.1 | 0.3 ±0.3 |
| 10 | 0.1 ±0.2 | 0.5 ±0.4 | 0.7 ±0.8 | 1.3 ±1.6 |
| 11 | 0.0 ±0.0 | 0.1 ±0.1 | 0.4 ±0.8 | 0.3 ±0.9 |
| 12 | 0.2 ±0.1 | 0.4 ±0.4 | 0.6 ±0.9 | 0.3 ±0.4 |
| 13 | 0.2 ±0.5 | 0.3 ±0.4 | 0.8 ±0.9 | 1.1 ±0.9 |
| 14 | 1.8 ±2.6 | 1.3 ±1.6 | 0.1 ±0.2 | 4.7 ±5.2 |
| 15 | 2.1 ±3.4 | 1.6 ±2.8 | 0.3 ±0.5 | 10.3 ±8.6 |
| 16 | 1.9 ±2.2 | 0.9 ±1.3 | 9.1 ±8.1 | 34.1 ±22.8 |
| 17 | 1.6 ±0.8 | 1.4 ±3.4 | 43.7 ±18.6 | 51.1 ±12.9 |
| 18 | 0.3 ±0.4 | 0.7 ±1.4 | 2.3 ±3.6 | 12.8 ±9.0 |
| 19 | 3.4 ±4.0 | 6.1 ±7.3 | 50.2 ±8.4 | 73.1 ±23.9 |
| 20 | 0.0 ±0.0 | 0.0 ±0.0 | 3.2 ±2.5 | 2.6 ±2.8 |
| avg | 0.73 ±0.89 | 1.12 ±1.62 | 6.34 ±3.32 | 11.21 ±6.65 |

## APPENDIX F   bAbI ATTENTION VISUALIZATION

We present a visualization of the attention distributions on bAbI tasks for a couple of examples. The visualization of attention weights is over different time steps based on different heads over all the facts in the story and a question. Different color bars on the left side indicate attention weights based on different heads (4 heads in total).

```
An example from tasks 1:    (requiring one supportive fact to solve)

Story:
                            John travelled to the hallway.
                            Mary journeyed to the bathroom.
                            Daniel went back to the bathroom.
                            John moved to the bedroom

Question:
                            Where is Mary?
Model's output:
                            bathroom
```

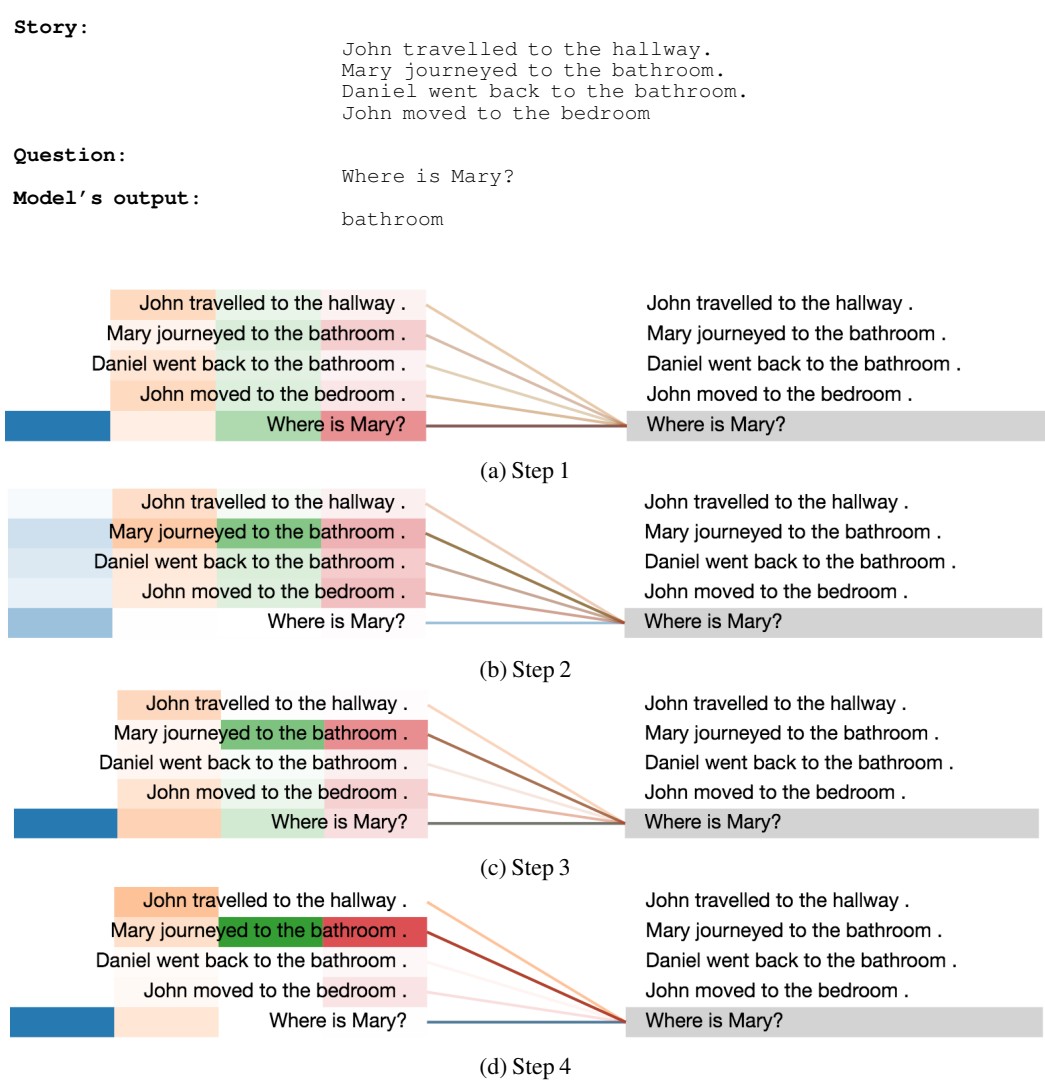

Figure 5: Visualization of the attention distributions, when encoding the question: *"Where is Mary?"*.

```
An example from tasks 2:    (requiring two supportive facts to solve)

Story:
                            Sandra journeyed to the hallway.
                            Mary went to the bathroom.
                            Mary took the apple there.
                            Mary dropped the apple.

Question:
                            Where is the apple?
Model's output:
                            bathroom
```

(a) Step 1

(b) Step 2

(c) Step 3

(d) Step 4

Figure 6: Visualization of the attention distributions, when encoding the question: *"Where is the apple?"*.

```
An example from tasks 2:    (requiring two supportive facts to solve)

Story:
                            John went to the hallway.
                            John went back to the bathroom.
                            John grabbed the milk there.
                            Sandra went back to the office.
                            Sandra journeyed to the kitchen.
                            Sandra got the apple there.
                            Sandra dropped the apple there.
                            John dropped the milk.

Question:
                            Where is the milk?
Model's output:

                            bathroom
```

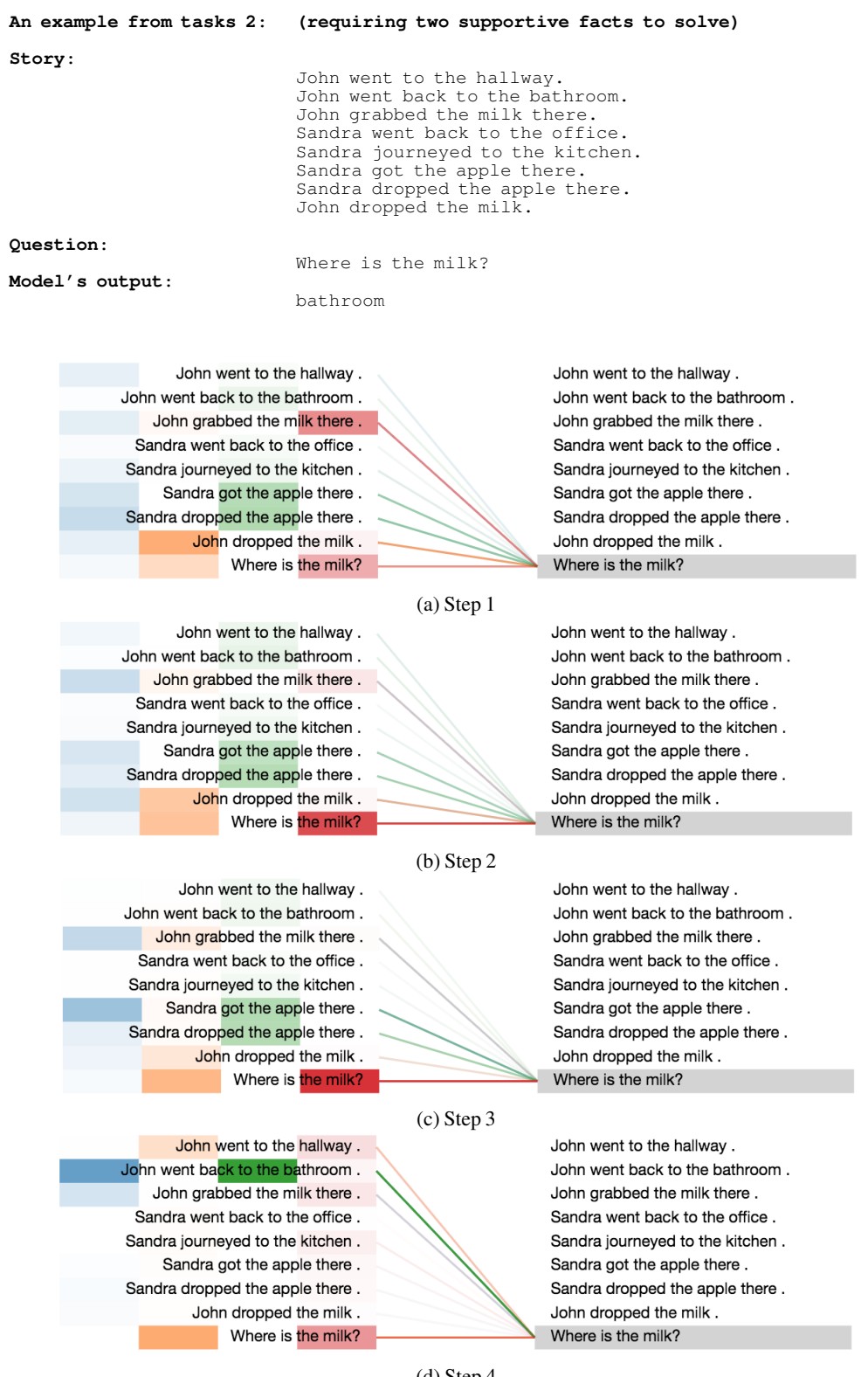

(a) Step 1

(b) Step 2

(c) Step 3

(d) Step 4

Figure 7: Visualization of the attention distributions, when encoding the question: *"Where is the milk?"*.

**An example from tasks 3:   (requiring three supportive facts to solve)**

**Story:**
Mary got the milk.

                        John moved to the bedroom.
                        Daniel journeyed to the office.
                        John grabbed the apple there.
                        John got the football.
                        John journeyed to the garden.
                        Mary left the milk.
                        John left the football.
                        Daniel moved to the garden.
                        Daniel grabbed the football.
                        Mary moved to the hallway.
                        Mary went to the kitchen.
                        John put down the apple there.
                        John picked up the apple.
                        Sandra moved to the hallway.
                        Daniel left the football there.
                        Daniel took the football.
                        John travelled to the kitchen.
                        Daniel dropped the football.
                        John dropped the apple.
                        John grabbed the apple.
                        John went to the office.
                        Sandra went back to the bedroom.
                        Sandra took the milk.
                        John journeyed to the bathroom.
                        John travelled to the office.
                        Sandra left the milk.
                        Mary went to the bedroom.
                        Mary moved to the office.
                        John travelled to the hallway.
                        Sandra moved to the garden.
                        Mary moved to the kitchen.
                        Daniel took the football.
                        Mary journeyed to the bedroom.
                        Mary grabbed the milk there.
                        Mary discarded the milk.
                        John went to the garden.
                        John discarded the apple there.

**Question:**
                        Where was the apple before the bathroom?

**Model's output:**
                        office

(e) Step 1

(f) Step 2

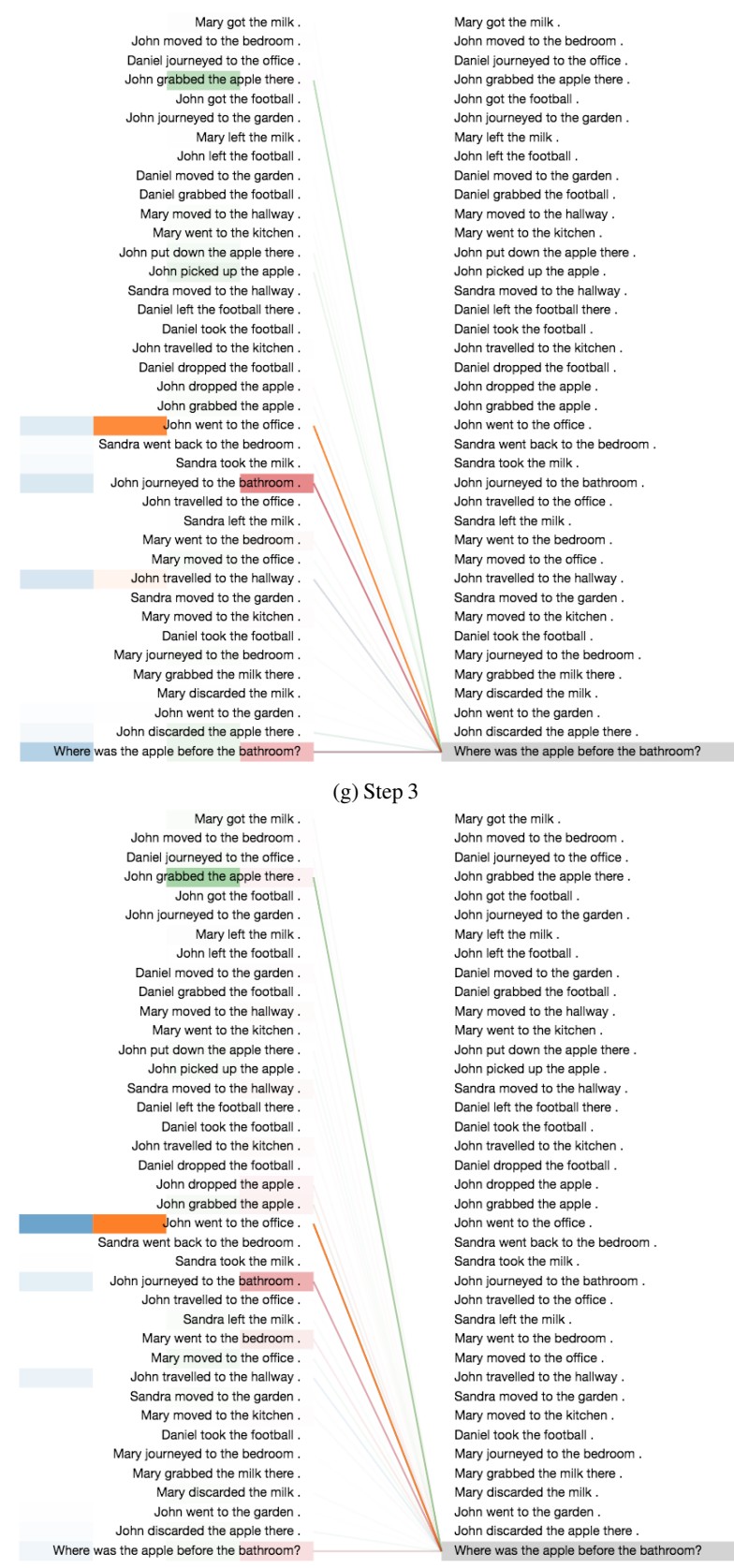

(g) Step 3

(h) Step 4

Figure 7: Visualization of the attention distributions, when encoding the question: *"Where was the apple before the bathroom?"*.

