# OpenReview forum: "Universal Transformers"
_ICLR.cc/2019/Conference_

### Official Review · AnonReviewer2 · 2018-11-02
**Good paper, contribution moderate, experiments promising**

**Rating:** 8
**Confidence:** 4

**Review:**

My summary: A new model, the UT, is based on the Transformer model, with added recurrence and dynamic halting of the recurrence. The UT should unite the computational universality properties of Neural Turing Machines and Neural GPU with good performance on disparate language and algorithmic tasks.

(I have read your author feedback and have modified my rating according to my understanding.)

Review:
The paper is well written and proofread, concrete and clear. The model is quite clearly explained, especially with the additional space of the supplementary material, appendices A and B (note fig 4 is less good quality than fig 2 for some reason) -- I’m fine with the use of the Supp Mat for this purpose.

The experiments have been conducted well, and demonstrate a wide range of tasks, which seems to suggest that the UT has pretty general purpose. The range of algorithmic tasks is limited, e.g. compared to the NTM paper.
I miss any experimental details at all on training.
I miss a comparison to Neural GPU and Stack RNN in 3.1, 3.2.

I miss a proof that the UT is computationally equivalent to a Turing machine. It does not have externally addressable, shared memory like a tape, and I’m not sure how to transpose read/write heads either.

The argument that the UT offers a good balance between inductive bias and expressivity is weak, though it may be the best one can hope for of a statistical model in a way. I note that in 3.1, the Transformer overfits, while it seems to underfit in 3.3 (lower LM and RC accuracy, higher LM perplexity), while the UT fare well, which suggests that the UT hits the balance better than the Transformer, at least.

From the point of view of network structure, it seems natural to lift further constraints on the model:
why should width of intermediate layers be exactly equal to sequence length?
why should all hidden state vectors be size $d$, the size of the embeddings chosen at the first layer, which might be chosen out of purely practical reasons like the availability of pre-trained word embeddings?

What is the contribution of this work? It starts from the Transformer, the ACT idea for dynamic halting in recurrent nets, the need for models fit for algorithmic tasks.
The UT’s building blocks are near-identical to the Transformers (and the paper is upfront and does a good job of explaining these similarities, fortunately)
- cf eq1-5: residuals, multi-headed self attention, and layer norm around all this.
- shared weights among all such units
- encoder-decoder architecture
- autoregressive decoder with teacher forcing
- decoder units like the encoder’s but with extra layer of attention to final output of encoder
- coordinate embeddings
The authors may correct me, but I believe that the UT with FC layers is exactly identical to the Transformer described in Vaswani 2017 for T=6.
So this paper introduces the idea of varying T, interprets it as a form of recurrence, and adds dynamic halting with ACT to that. Interestingly, the recurrence is not over sequence positions here.
This contribution is not major, on the other hand the experimental validation suggests the model is promising.

Typos and writing suggestions
above eq 8: masked such that -> masked so that
eq 8: dimensions of O and H^T are incompatible: d*V, m*d; to evacuate the notation issue for transposition, cf footnote 1, here and elsewhere, you could use either ${^t A}$ or $A^\top$ or $A^\intercal$. You could also write $t=T$ instead of just $T$.
sec3.3 line -1: designed such that -> designed so that
Towards the beginning of the paper, it may be useful to stabilise terminology for $t$: depth (as opposed to width for $m$), time steps, recurrence dimension, revisions, refinements

---

> ### Author Response · Authors · 2018-11-27
> **Rebuttal Part 2**
>
> >> why should width of intermediate layers be exactly equal to sequence length?
>
> If we understand correctly, the question is “Why only have one vector per input symbol at every intermediate layer/step?”. With the self-attention mechanism, both in Transformer and the Universal Transformer at each layer/step, we revise the representation of each symbol given the representations of all the other input symbols in the previous layer/step. Thus, we need vectors representing each symbol in the input at each intermediate layer/step (illustrated in Fig. 1 in the paper).
>
> >> why should all hidden state vectors be size $d$, the size of the embeddings chosen at the first layer, which might be chosen out of purely practical reasons like the availability of pre-trained word embeddings?
>
> Indeed, there is no architectural constraint in UT for having the same size for the hidden state and input/output embeddings (same as with standard Transformer). These are independent hyper-parameters and one can set different values for them, although this has not really been done in any other transformer-based work as far as we are aware.
>
> >>The authors may correct me, but I believe that the UT with FC layers is exactly identical to the Transformer described in Vaswani 2017 for T=6.
>
> No, there are several differences (which prove to be important theoretically and in practice):
>
> * In UT, parameters are tied across layers (i.e. the same self-attention and the same transition function is applied across recurrent steps); Transformer has different weights for each layer / step. This is important because a UT trained on T=4 steps can be evaluated using any T, whereas a Transformer trained with T layers/steps can only be evaluated for the same T steps.
> * Besides the position embedding, we also have time-step embeddings, which are combined into (essentially 2-D) “coordinate embeddings”
> * We introduce the coordinate embedding at the beginning of each step (not just once at t_0)
> * Lastly, ACT makes T dynamic for each position, whereas with Transformer T is static.
>
> >> So this paper introduces the idea of varying T, interprets it as a form of recurrence, and adds dynamic halting with ACT to that. Interestingly, the recurrence is not over sequence positions here.
>
> It is in fact the other way around: We introduce recurrence over processing steps (by sharing/tying the transition weights), and that allows us to vary T. We then add ACT to that.
>
> (As noted above: You cannot vary T / number of layers between training and testing in a standard Transformer as it is trained with a different set of weights for each of the T layers.)
>
> >>Typos and writing suggestions
> Thanks, we’ve updated these in the revised version. We also increased the resolution of the image in the Figure 4.

---

> > ### Comment · AnonReviewer2 · 2018-11-27
> > **Worth emphasising: difference with Transformer: tying across recurrence, different train/ test depth T**
> >
> > Thanks for your feedback.
> > Regarding the following argument:
> > >>> * In UT, parameters are tied across layers (i.e. the same self-attention and the same transition function is applied across recurrent steps); Transformer has different weights for each layer / step. This is important because a UT trained on T=4 steps can be evaluated using any T, whereas a Transformer trained with T layers/steps can only be evaluated for the same T steps.
> > I guess I had understood this but had not realised the implications. To make the paper persuasive, it might be worth emphasising this specific point.

---

> ### Author Response · Authors · 2018-11-27
> **Rebuttal Part 1**
>
> We thank the reviewer for the thorough review, and respond below. We have also updated the paper to address these comments.
>
> >> “What is the contribution of this work [...]”
>
> We introduce two changes to the Transformer architecture (namely adding recurrence and dynamic computation) which:
>
> 1) increase the model’s theoretical capabilities (make it Turing-complete),
> 2) significantly improve results (compared to standard Transformer) on all tasks that it was evaluated on including large-scale MT (UT improves over standard Transformer by 0.9 BLEU on WMT14 En-De), and lastly
> 3) also increase the *types* of tasks Transformer can learn in the first place (eg a standard Transformer fails on bAbI (solves only 50% of tasks; see Table 1), is vastly outperformed by LSTMs on subject-verb agreement (Table 2), and achieves a test perplexity of 7,321 on LAMBADA (Table 3); on the other hand UT solves 100% of bAbI tasks, outperforms LSTMs on SVA prediction, even performing progressively better as the number of attractors increases, and achieves a state-of-the-art test perplexity of 142 on LAMBADA).
>
> While we agree (and readily point out throughout) that these are two fairly simple architectural changes, we do want to point out that this yields a new type of parallel-in-time recurrent self-attentive model which blends the best of both worlds of RNNs and Transformers, is theoretically superior to standard Transformers, and practically leads to vastly improved results across a much wider range of tasks, as mentioned above.
>
> >> Range of algorithmic tasks limited; experimental / training details missing
> The main purpose of evaluating our model on algorithmic tasks is to probe its ability for length generalization in a controlled setup, where we train on 40 symbols and test on 400 symbols. We intentionally chose three simple tasks, i.e. copy, reverse, and addition to mainly focus on the length generalization aspect of the problem, and as can be seen, Transformers and LSTMs perform poorly in this setup in terms of sequence accuracy, while UT is doing a much better job (despite the fact that it’s not trained with a custom curriculum learning like Neural GPU to perform well on these tasks). Furthermore, we also tested our model on Learning-to-Execute tasks which can be considered in the family of algorithmic tasks.
>
> We have added additional experimental and training details to the revised version of the paper.
>
> >> I miss a comparison to Neural GPU and Stack RNN in 3.1 and 3.2
>
> This is because for each of the tasks we only reported the state-of-the-art / best performing baselines and Neural GPUs and Stack RNNs have been outperformed by other methods for both bAbI (3.1) and subject-verb agreement prediction (3.2).
>
> >> I miss a proof that the UT is computationally equivalent to a Turing machine. It does not have externally addressable, shared memory like a tape, and I’m not sure how to transpose read/write heads either.
>
> The proof included in the paper goes by reduction from the Neural GPU which in turn goes by reduction from cellular automata. So this line of proof does not operate directly on a tape or read/write heads, it starts from cellular automatas’ universality (like the game of life). We have also added an Appendix B to elaborate on this with an example.

---

### Official Review · AnonReviewer1 · 2018-11-02
**Solid and empirically promising model which merges Transformer and recurrent models but without strong intuitive or theoretical support to back up its claims.**

**Rating:** 6
**Confidence:** 2

**Review:**

This paper describes a transformer with recurrent structure to take advantage of self-attention mechanism. The number of recurrences can be dynamically determined through ACT-like halting depending on the difficulty of the input. A series of experiments on language modeling tasks have been demonstrated to show promising performances.

The overall concerns about this paper is that while the performances are quite promising, the theoretical claims and comparisons in the discussion section are of question. The authors attempt to provide connections to other networks (i.e., Natural GPU, RNN) since UT is an amalgamation of both transformers and RNN, they sound a little “hand-wavy” (i.e., comments about UT effectively interpolating between the feed-forward, fixed-depth Transformer and a gated recurrent architecture). In short, while empirically completely acceptable, intuitively or theoretically it is hard to grasp why UT is superior other than the dynamic/sharing layers across t (not time). I believe that improving this aspect could make this paper even better. Based on the comments below and the responses with the authors, I am willing to improve my score.

Pros:
1.	The best of both worlds from parallelizable transformer and recurrent structure for repeated self-attention mechanism. Essentially, the “depth” of the transformer can vary if we “unroll” the recurrent stacks.

2.	Extensive experiments showing the performance of UT.

3.	Analysis of the effect of the recurrent aspect of UT and how it can vary depending on the task difficulty.

Comments/cons:
1.	I am having trouble understanding the “universal” aspect of the transformer. Is this because the variability of the depth of UT (since “given sufficient memory” was mentioned)? If so, such characteristic of “computational universality” does not seem much unique to UT compared to infinite memory for a transformer or a simple RNN across stack (i.e., input is the while sequence and recurrent step is through the stack analogous to UT stack). Please comment on this.

2.	It is nice to see many experiments, but without preexisting knowledge about the datasets and their tasks, I can only make relative judgements based on the provided comparisons against other methods. It would be nice to see slightly more detailed descriptions of each task (particularly LAMBADA LM), not necessarily in the main paper (due to space) but in the appendix if possible for improved self-containedness.

3.	In the discussion, the crucial difference between UT and RNN is that RNN is stated to be that RNN cannot access memory in the recurrent steps while UT can. This seems to be the case for not just UT but any Transformer-type model by construction.

4.	The authors stated that the “recurrent step” for RNN is through time (as the authors stated) while the “recurrent step” in UT is not through time. While this claim is completely correct itself, the RNN’s inability to access memory in its “recurrent steps” was compared with how UT could still access memory throughout its “recurrent steps”. In this sense, we may argue that the UT cannot access memory across its own t (stacking across t). I am not sure if it is fair to make such implications by putting both “recurrent steps” to be of same nature and pointing out one’s weakness. Perhaps the authors could comment on this.

Minor:
1.	Table 2.: Best Stack-RNN for 1 attractor is the highest but not bold-faced.

---

> ### Author Response · Authors · 2018-11-27
> **Rebuttal**
>
> We thank the reviewer for the thorough review, and respond below. We have also updated the paper to address these comments.
>
> >> Questions around Universality of UT
>
> The main ingredient for the universality of UT comes from the recurrence in depth. Unbounded memory is also important, but it’s the sharing of weights combined with adaptive computation time that brings universality -- even with unbounded size, the standard Transformer would not be universal. We have added an Appendix B to elaborate on this with an illustrative example.
>
> >> More detailed descriptions of the tasks
>
> We’ve added an appendix D, which provides more detail on the tasks and datasets.
>
> >> 3. In the discussion, the crucial difference between UT and RNN is that RNN is stated to be that RNN cannot access memory in the recurrent steps while UT can. This seems to be the case for not just UT but any Transformer-type model by construction.
>
> This is correct in the sense that UT, like transformer, can access memory in each of its processing steps. But the crucial difference is that UT, unlike transformer, is recurrent in its steps (similar to RNNs), where the standard Transformer is like a deep feed-forward model where each step is computed using a separate, learned layer. So, as we stated in the paper, “UTs combine the  parallelizability and global receptive field (access to the memory) of feed-forward sequence models like the Transformer with the recurrent inductive bias of RNNs”. As the experiments demonstrate, this *combination* yields very strong results across a wider range of tasks than either on its own.
>
> >> 4. The authors stated that the “recurrent step” for RNN is through time (as the authors stated) while the “recurrent step” in UT is not through time. [...] In this sense, we may argue that the UT cannot access memory across its own t (stacking across t). [...]
>
> Yes, this is a good point and indeed correct in terms of the model as reported in the paper. We did also implement a variant of UT where in every step (in depth “t”) the model attends to the output of all the previous steps (not just the last one; i.e. it has access to memory across t), but it didn’t improve results in our experiments. We speculate that this may be because being able to access memory in time (i.e. across sequence length), in particular for language tasks, is more important than being able to access all the previous transformations (i.e. access memory in depth).
>
> Furthermore, we also note that the maximum number of steps in depth (denoted $T$ in the paper) is typically *much fewer* than the maximum length of the sequences (denoted $m$ in the paper). This makes access to previous transformations less useful across "recurrent steps" for UTs as the recurrence allows the model to memorize its transformations across the shorter paths in depth (due to vanishing gradient playing a smaller role), and so being able to look up memory in each step (“across its own t” as the reviewer mentions) therefore becomes less useful.

---

> ### Public Comment · (anonymous) · 2018-12-12
> **Wrong claim in this paper**
>
> The claim stated in this paper "Transformers are not Turing-complete" is wrong. It's proved in [1] that Transformer is Turing-complete.
>
> [1] https://openreview.net/forum?id=HyGBdo0qFm&noteId=HyGBdo0qFm

---

> > ### Author Response · Authors · 2018-12-12
> > **"Transformer with positional encodings and fixed precision is not Turing complete", from [1]**
> >
> > This is incorrect. Please see our response to the same comment with the heading "Potentially wrong claim in this paper".

---

### Official Review · AnonReviewer3 · 2018-11-03
**Recursively applying multihead self-attention block in Transformer, small change leads to effective improvements on multiple tasks.**

**Rating:** 6
**Confidence:** 4

**Review:**

This paper extends Transformer by recursively applying a multi-head self-attention block, rather than stack multiple blocks in the vanilla Transformer. An extra transition function is applied between the recursive blocks. This combines the idea from RNN and attention-based models. But the RNN structure here is not applied to the input sequence, but to the sequence of blocks inside the Transformer encoder/decoder. In addition, it also uses a dynamic adaptive computation time (ACT) halting mechanism on each position, as suggested by the previous ACT paper. In fact, it can be seen as a memory network with a dynamic number of hops at the symbol level.

The paper is well-written and easy to follow. The experimental results demonstrate that the proposed model can achieve state-of-the-art prediction quality in several algorithmic and NLP tasks.

Pros
1. The proposed UT is compatible with both algorithmic and NLP tasks by combining the Transformer with weight sharing of recurrence and dynamic halting. In contrast, previous algorithmic and NLP takes can only be solved by more specific neural architectures (e.g., NTM for algorithmic tasks and the Transformer for NLP tasks).
2. The empirical results verify the effectiveness of the UT on several benchmarks.
3. The careful experimental analyses not only show the insight of dynamic halting in QA task but demonstrate the ACT is very useful for algorithmic tasks.
4. The publicly-released codes could make great contributions to the NLP community.

Cons
1. It proposes an incremental change to the original Transformer by introducing recursive connection between multihead self-attention blocks with ACT. The idea behind UT is similar to memory networks and multi-hop reasoning.
2. The recursive structure is not applied to the input sequence, so UT does not have the advantage of RNN/LSTM on capturing sequential information and high-order features.
3. Although evaluated on multiple datasets and tasks, they only cover simple QA task and EN-DE translation task. Comparing to other papers applying modifications to Transformer, it is better to include at least one heavy task on large/challenging dataset/task.
4. On machine translation task, why does the model without dynamic halting achieve the SOTA performance? This is in contrast to the claim of the advantage of using dynamic halting.
5. The ablation studies focus only on the dynamic halting, but what if weight sharing is removed from the UT?

---

> ### Author Response · Authors · 2018-11-27
> **Rebuttal Part 2**
>
> >>3. Although evaluated on multiple datasets and tasks, they only cover simple QA task and EN-DE translation task. Comparing to other papers applying modifications to Transformer, it is better to include at least one heavy task on large/challenging dataset/task.
>
> We chose an array of 6 different tasks (ranging from smaller and more structured, to large-scale in the case of the WMT machine translation experiments) in order to measure and highlight different capabilities of UT compared to other models:
>
> * We chose bAbI-QA since its set of 20 different tasks each tests a unique aspect of language understanding and reasoning. Besides this, the bAbI-1k data set (as opposed to the 10k version) is quite a challenging setup since a model should be very data efficient to be able to get reasonable results on this data, and as we show, the Transformer (and LSTMs for that matter) are *not* able to solve these tasks. Therefore, given that these state-of-the-art sequence models fail here, we believe evaluating on these tasks to be a reasonable first step to benchmark the capabilities of UTs against other models on (admittedly simpler) structured linguistic inference tasks.
> * Algorithmic tasks and LTE tasks are also considered as a set of controlled experiments that first of all helps us to compare the model with other theoretically-appealing models like Neural GPU, and to test the models in terms of some specific aspects such as length-generalization or ability to model nesting in the input source code (where again, LSTMs and the Transformer perform very poorly).
> * The subject-verb agreement task is chosen as it has been shown [1] that the lack of recurrence can prevent the Transformer from solving this task, whereas we show that the Universal Transformer easily solves it and in fact improves as the task gets harder, i.e. more attractors are introduced (last paragraph, Sec 3.2).
> * Lambada is a challenging large-scale dataset which highlights the difficulties of incorporating broader context in the task of language modeling. Achieving SOTA on this dataset is further evidence that the Universal Transformer provides a better inductive bias for language understanding.
> * And finally, experiments on the large-scale machine translation task, WMT2014-ENDE, show that the Universal Transformer is not only a theoretically-appealing model, but also a model that performs well on practical real-world tasks.
>
> We believe that, together, this set of 6 diverse tasks highlights the different strengths and weaknesses of UT, especially compared to the well established LSTM and Transformer baselines, and we leave more investigation with more datasets/tasks for future studies.
> ---------------------------------------------------------------
> [1] Tran, Ke, Arianna Bisazza, and Christof Monz. "The Importance of Being Recurrent for Modeling Hierarchical Structure." arXiv preprint arXiv:1803.03585 (2018).
>
>
> >>4. On machine translation task, why does the model without dynamic halting achieve the SOTA performance? This is in contrast to the claim of the advantage of using dynamic halting.
>
> The advantage of dynamic halting is that it mainly helps in the smaller (bAbI, SVA) and more structured tasks (Lambada). On MT we achieved marginally better results without it. We believe this is because dynamic halting acts as a useful regularizer on the smaller tasks, and is therefore not as useful when more data is available in the large-scale MT task. We mention this in the discussion of our results, but we emphasize this even more in the revised version of the Introduction.
>
> >> 5. The ablation studies focus only on the dynamic halting, but what if weight sharing is removed from the UT?
>
> As noted above, UT without weight-sharing (across depth) is not recurrent (as separate transition functions are learned for each step/”layer”), so it cannot generate a variable number of revisions / processing steps, and therefore also cannot use dynamic halting. It is only with shared transition blocks that the model becomes recurrent, allowing the use of dynamic halting / ACT.

---

> ### Author Response · Authors · 2018-11-27
> **Rebuttal Part 1**
>
> We thank the reviewer for the thorough review and respond below. We have also updated the paper to address these comments.
>
> >>extends Transformer by recursively applying a multi-head self-attention block, rather than stack multiple blocks in the vanilla Transformer. An extra transition function is applied between the recursive blocks
>
> To avoid any potential confusion about the architecture, we note that the {multi-head self-attention + transition}-block is applied recursively *as a whole*. The Transition function is not “extra”, it also exists in the standard Transformer, but the difference is that we apply the same Transition function at every layer / step (by tying the weights). This makes the model recurrent (in “depth” or in its concurrent processing steps), which then allows us to vary the number of steps and add dynamic halting -- both impossible with the standard Transformer architecture.
>
>
> >>it also uses a dynamic adaptive computation time (ACT) halting mechanism on each position, as suggested by the previous ACT paper
>
> ACT was introduced and applied in the context of a sequential RNN model where each symbol is processed one after the other, but with a variable number of steps each. However we apply ACT concurrently to all symbols (i.e. in a parallel-in-time model). It has the same effect of allowing a variable number of processing steps per symbol, but we want to emphasize that the way it is used in UT is different from the original ACT paper (in depth vs in sequence length / time).
>
> >>1. [...] The idea behind UT is similar to memory networks and multi-hop reasoning.
>
> Yes, indeed, the idea behind UT is related to memory networks. We mentioned this briefly (last paragraph of Section 4), but have expanded on this in the updated version: In UT, similar to dynamic memory networks, there is an iterative attention process which allows the model to condition its attention over memory on the result of previous iterations.  As we also show in the visualization of the attention distributions for the bAbI task (Appendix F in the revised paper), we can see that there is a notion of temporal states in UT,  where the model updates the memory (states) in each step based on the output of previous steps, and this chain of updates can indeed be viewed as steps in a multi-hop reasoning process.
>
> >>2. The recursive structure is not applied to the input sequence, so UT does not have the advantage of RNN/LSTM on capturing sequential information and high-order features.
>
> We disagree with this statement: In self-attentive parallel-in-time models (such as Transformer or UT) information is exchanged between symbols (i.e. sequential information) using the self-attention mechanism. Therefore, in the first step each symbol representation is already conditioned on every other symbol (i.e. includes first-order features). However, as this process is continued, with each additional processing step UTs are in fact able to capture higher-order features between symbols.

---

### Public Comment · ~Zihao_Ye1 · 2018-11-26
**Probably a typo?**

In eq 4, you wrote $H^t=LayerNorm(A^{t-1}+Transition(A^t))$.
But according to your text description and figure 4, I suppose it should be $H^t=LayerNorm(A^t +Transition(A^t))$, otherwise, there would be a cross-step residual connection which is not mentioned in the paper.

---

> ### Author Response · Authors · 2018-11-26
> **Eq 4 is $H^t=LayerNorm(A^t +Transition(A^t))$ in our submission**
>
> Thanks for the comment.  If you download and check the pdf of our submission in OpenReview, equation 4 is in fact $H^t=LayerNorm(A^t +Transition(A^t))$, and not $H^t=LayerNorm(A^{t-1}+Transition(A^t))$.
>
> There is, however, a small typo in eqn 5. It should be $A^t =LAYERNORM((H^{t−1}+P^t ))+MULTIHEADSELFATTENTION(H^{t−1}+P^t ))$ instead of  $A^t =LAYERNORM(H^{t−1}+MULTIHEADSELFATTENTION(H^{t−1}+P^t ))$, as the residual connection in our model adds up the input "with coordinate embedding" to the state. We already fixed this in the revised version of our submission and will upload it to OpenReview soon.

---

> > ### Public Comment · ~Zihao_Ye1 · 2018-11-26
> > **Thanks for your reply**
> >
> > oops, the typo I mentioned exist in your arXiv submission rather than openreview submission, sorry about the mistake.
> > Also thanks for your notice about eqn 5.

---

### Public Comment · (anonymous) · 2018-12-12
**Potentially wrong claim in this paper**

The claim stated in this paper "Transformers are not Turing-complete" is potentially wrong. It's proved in [1] that Transformer is Turing-complete. It is definitely necessary to address this concern before this paper can be accepted.

[1] https://openreview.net/forum?id=HyGBdo0qFm&noteId=HyGBdo0qFm

---

> ### Author Response · Authors · 2018-12-12
> **Transformer with fixed-precision is not Turing-complete**
>
> Thanks for your comment.
>
> The main point here is that in [1] the authors assume arbitrary-precision arithmetic, as clarified in their responses on OpenReview where they noted "Our proofs are based on having unbounded precision for internal representations [...]". Therefore, as mentioned in their section "The need of arbitrary precision", "[...] the Transformer with positional encodings and fixed precision is not Turing complete." In other words, in practice (i.e. assuming fixed-precision arithmetic), the Transformer is *not* computationally universal.
>
> To see this, note that in fixed-precision arithmetic a single multiply is O(1) (and so are the nonlinearities). Therefore the computation of the fixed number of attention layers in the Transformer is at most O(n^2), which is polynomial time, while there exist computable functions that are not computed in polynomial time. Or stated in another way: If a model only has a specific time-window, like O(n^2), there are problems it cannot solve, hence it cannot be universal (see also [2] for more on this).
>
> In the Universal Transformer, on the other hand, this time-window is *not* fixed (see Appendix B in the revised version of our paper on OpenReview for an intuitive example). As pointed out by AnonReviewer2 below, we further want to emphasize that this is because the recurrence resulting from tying the weights allows one to vary the number of time-steps T arbitrarily at inference time (i.e. you can train with T=4 and test with any T). This potentially unbounded time-window (which is only possible because of its recurrence) is what makes UT computationally universal.
>
> We will clarify these points in the revised version of the paper.
>
> ----
> [1] https://openreview.net/forum?id=HyGBdo0qFm&noteId=HyGBdo0qFm
> [2] https://en.wikipedia.org/wiki/Time_hierarchy_theorem

---

### Meta-Review · Area_Chair1 · 2018-12-14
**Universal Transformers (with optional dynamic halting/ACT)**

**Confidence:** 4
**Recommendation:** Accept (Poster)

**Metareview:**

This paper presents Universal Transformers that generalizes Transformers with recurrent connections. The goal of Universal Transformers is to combine the strength of feed-forward convolutional architectures (parallelizability and global receptive fields) with the strength of recurrent neural networks (sequential inductive bias). In addition, the paper investigates a dynamic halting scheme (by adapting Adaptive Computation Time (ACT) of Graves 2016) to allow each individual subsequence to stop recurrent computation dynamically.

Pros:
The paper presents a new generalized architecture that brings a reasonable novelty over the previous Transformers when combined with the dynamic halting scheme. Empirical results are reasonably comprehensive and the codebase is publicly available.

Cons:
Unlike RNNs, the network recurs T times over the entire sequence of length M, thus it is not a literal combination of Transformers with RNNs, but only inspired by RNNs. Thus the proposed architecture does not precisely replicate the sequential inductive bias of RNNs. Furthermore, depending on how one views it, the network architecture is not entirely novel in that it is reminiscent of the previous memory network extensions with multi-hop reasoning (--- a point raised by R1 and R2). While several datasets are covered in the empirical study, the selected datasets may be biased toward simpler/easier tasks (--- R1).

Verdict:
While key ideas might not be entirely novel (R1/R2), the novelty comes from the fact that these ideas have not been combined and experimented in this exact form of Universal Transformers (with optional dynamic halting/ACT), and that the empirical results are reasonably broad and strong, while not entirely impressive (R1). Sufficient novelty and substance overall, and no issues that are dealbreakers.